# Guaranteed Trust Region Optimization via Two-Phase KL Penalization

## Abstract

On-policy reinforcement learning (RL) has become a popular framework for solving sequential decision problems due to its computational efficiency and theoretical simplicity. Some on-policy methods guarantee every policy update is constrained to a trust region relative to the prior policy to ensure training stability. These methods often require computationally intensive non-linear optimization or require a particular form of action distribution. In this work, we show that applying KL penalization alone is nearly sufficient to enforce such trust regions. Then, we show that introducing a "fixup" phase is sufficient to guarantee a trust region is enforced on *every* policy update while adding fewer than 5% additional gradient steps in practice. The resulting algorithm, which we call FixPO, is able to train a variety of policy architectures and action spaces, is easy to implement, and produces results competitive with other trust region methods.

## 1 Introduction

On-policy reinforcement learning (RL) methods seek to optimize a stochastic policy, where a neural network is used to parameterize a distribution $\pi(a|s)$ over actions conditioned on the current state. In this framework, most on-policy RL methods seek to limit the scale of updates between successive policies during optimization. Some on-policy RL methods operate by guaranteeing that each policy update remains within a "trust region" (Schulman et al., 2015a). These methods are used when training stability during a long period of training is essential. However, finding a policy update near the edge of the trust region often comes at significant computational cost. Another branch of on-policy methods instead perform "proximal" policy updates, that limit the *expected* scale of policy updates, but can result in individual policy updates being of arbitrary magnitude (Schulman et al., 2017a). These methods are much more computationally efficient, but large-scale training can require the use of multiple training runs or human intervention to recover from training instabilities. In this work we propose Fixup Policy Optimization (FixPO), which combines both a proximal primary phase with a precise fixup phase, that operate by sharing a single penalty coefficient $\beta$. By performing a more conservative proximal update before strictly enforcing a trust region, FixPO is able to approximately match the computational efficiency and rewards of proximal methods while providing the same stability guarantees as trust region methods.

An important result in the development of trust-region methods is a proof presented with the Trust Region Policy Optimization algorithm (TRPO) (Schulman et al., 2015a) that for a particular value of $C$, iteratively applying the following update provably results in monotonic improvement of the expected return of $\pi_i$ :

$$\pi_{i+1} = \text{argmax}_\pi \left[ L_{\pi_i}(\pi) - C D_{KL}^{max}(\pi_i, \pi) \right] \qquad (1)$$

where $L_{\pi_i}$ is the importance sampled "policy gradient" loss, $D_{KL}^{max}(\pi_i, \pi)$ is the maximal value of the Kullback-Leibler (KL) divergence between the action distributions of $\pi_i(s|a)$ and of $\pi(s|a)$, and $C$ is a function of the characteristics of the Markov Decision Process (MDP). In practice, TRPO uses constrained optimization to perform policy updates subject to a constraint on the average KL divergence instead of only penalizing the maximal value. Due to constrained optimization preventing the use of minibatching and increasing the computational cost of optimizing deep neural networks, Proximal Policy Optimization algorithms (Schulman et al., 2017a) are more frequently used in practice. These methods do not guarantee that any precise trust region constraint is enforced but approximately limit the scale of $D_{KL}(\pi_i, \pi)$ .

The most well-studied PPO algorithm, often referred to as PPO-clip, or shortened to just PPO, operates by zeroing the loss contribution from likelihood ratios outside of the range $1 \pm \epsilon_{CLIP}$. Clipping in this way is very computationally efficient and ensures that for each state, at most one gradient step is taken which could increase the $D_{KL}(\pi_i, \pi)$ beyond the trust region. However, another class of PPO algorithms also introduced in Schulman et al. (2017a) instead feature a policy update inspired by the theory above:

$$\pi_{i+1} = \text{argmax}_\pi \left[ L_{\pi_i}(\pi) - \beta D_{KL}(\pi_i, \pi) \right] \tag{2}$$

In the above equation, $\beta$ is a hyperparameter that is typically tuned dynamically in response to the scale of recent policy updates as measured in $D_{KL}(\pi_i, \pi)$. Although this PPO variant is believed to perform worse than PPO-clip, its simplicity and connection to the above theory have made it a subject of study in several later works, which have extended it in various ways.

In this work, we demonstrate that by rapidly adapting $\beta$, it is possible to nearly enforce a trust region. Then, by performing a small number of additional gradient steps in a "fixup phase," we can guarantee the trust region is precisely enforced for a wide range of policy classes.

This work provides the following contributions:

1. An RL algorithm, FixPO, that efficiently enforces a guaranteed trust region between every policy update using only KL penalization.
2. Experiments showing the performance of the proposed algorithm on a variety of benchmarks compared to other trust region methods.
3. Ablation experiments showing the effect of each component of the proposed algorithm and why those components are necessary.

## 2 RELATED WORK

**Trust Region Methods**  The algorithm presented in this work follows in large part from the theory of trust region reinforcement learning methods, namely Schulman et al. (2015a) and Schulman et al. (2017a), combined with more recent insights from Andrychowicz et al. (2020). Work on FixPO was also guided by publications on PPO variants, such as Cobbe et al. (2020), from which the term "phase" was borrowed, and Hilton et al. (2021), which analyzes the effect of $\beta$ in relation to batch size. Works that analyze various aspects of PPO were also extremely useful, including Engstrom et al. (2020), which provides a detailed analysis of the relationship between PPO and TRPO, and Hsu et al. (2020), which examines several aspects of PPO in detail, such as the action distribution parameterization and effect of different KL penalties. More recently, Huang et al. (2022) provides an analysis of the effect of many proposed changes to PPO which was invaluable in this research.

Besides Schulman et al. (2015a), other methods for guaranteeing constrained updates have been proposed specifically for Gaussian policies (Akrour et al., 2019; Otto et al., 2021).

**Lagrangian Methods**  Although we are not aware of any comprehensive survey on the topic, loss functions structured similarly to Augmented Lagrangian methods (Hestenes, 1969) are frequently used in various Deep RL methods, including Song et al. (2019); Andrychowicz et al. (2020). Our proposed $L_\beta$ is similar to the losses proposed in those works, with two additions we describe in Section 3.1. Lagrangian methods used in some off-policy Deep RL work, such as for automatic entropy tuning in Haarnoja et al. (2018) and constraining offline Q estimates in Kumar et al. (2020). There are several applications of Lagrangian methods in Safe Deep RL works (Chow et al., 2015; Achiam et al., 2017), Imitation Learning and Inverse RL (Peng et al., 2018), Differentiable Economics (Dütting et al., 2017; Ivanov et al., 2022), and Multi-Agent RL (Ivanov et al., 2023).

**KL Regularized RL**  Outsides of trust region methods, using the KL divergence to regularize RL has been a long-standing method (Rawlik et al., 2012), and continues to be used in recent methods such as Kozuno et al. (2022), Vieillard et al. (2020), and Galashov et al. (2019). KL regularization is also a critical component of several recent offline RL methods, such as Wu et al. (2019), Nair et al. (2020), and Jaques et al. (2019).

**Benchmarks and Libraries**  The primary benchmarks used in this work were the Mujoco (Todorov et al., 2012) benchmarks from OpenAI Gym (Brockman et al., 2016), and the Meta-World (Yu et al., 2019) benchmarks. In most of our experiments, we make use of code from `Tianshou` (Weng et al., 2022), although we used `stable-baselines3` (Raffin et al., 2021) in earlier experiments. We also used `sample-factory` (Petrenko et al., 2020) to run experiments on tasks from the DMLab-30 (Beattie et al., 2016) benchmark.

## 3 METHOD

### 3.1 LOSS FUNCTIONS

Our method begins with the well-known loss function that results in policy updates that approximate Equation 2, also known as the KL regularized importance sampled policy gradient.

$$L(s, a, \hat{A}) = \underbrace{-\frac{\pi_\theta(a|s)}{\pi_{\theta'}(a|s)}\hat{A}}_{L_\pi} + \beta \overbrace{D_{KL}\left[\pi_\theta(a|s), \pi_{\theta'}(a|s)\right]}^{L_{KL}} \tag{3}$$

Where $\pi_\theta$ is the policy undergoing the update, $\pi_{\theta'}$ is the policy from the previous step, and $\hat{A}$ are advantages estimated using GAE (Schulman et al., 2015b). In order to define modified forms of this loss function, we also define the individual components, $L_\pi$ and $L_{KL}$, both of which will be used to optimize the policy parameters $\theta$.

We depart from Schulman et al. (2017a) in how we acquire $\beta$. Instead of using a fixed value or dynamically tuning $\beta$ on an epoch-by-epoch manner, we instead tune $\beta$ as a Lagrange multiplier using a loss similar to those described in Song et al. (2019); Andrychowicz et al. (2020). However, we make two modifications that differ from the losses described in those works. First, we enforce a target on $D_{KL}^{max}\left[\pi_\theta, \pi_{\theta'}\right]$, which is theoretically justified by Equation 2. Although typically dismissed as fragile to outliers, we find that the maximal KL value is less sensitive to hyperparameter choices, which we discuss in Section 4.1. Secondly, we add a term $C_\beta$, which mirrors the $C$ value in Equation 2. This results in moving the optima of the Lagrangian optimization away from the constraint surface, which we discuss in more detail in Paragraph 15. This results in the following loss, which tunes $\beta$ to approximately enforce the trust region constraint.

$$L_\beta = \beta \,\mathbf{sg}\left[\epsilon_{KL} - C_\beta D_{KL}^{max}\left[\pi_\theta, \pi_{\theta'}\right]\right] \tag{4}$$

Where $\mathbf{sg}$ is the "stop-gradient" operator, which we include as a reminder that this loss function should only be tuning $\beta$, and should not modify the policy parameters $\theta$. $\epsilon_{KL}$ is a hyperparameter that controls the size of the trust region, which performs a similar role as $\epsilon_{CLIP}$ in PPO-clip. $C_\beta$ moves the target of the primary phase away from the edge of the trust region and compensates for bias introduced by computing $D_{KL}^{max}$ on minibatches. When $C_\beta = 1$, this loss function tunes $\beta$ such that $D_{KL}^{max} \approx \epsilon_{KL}$, by increasing $\beta$ when $C_\beta D_{KL}^{max} > \epsilon_{KL}$ and decreasing $\beta$ when $C_\beta D_{KL}^{max} < \epsilon_{KL}$. When $C_\beta > 1$, optimizing $L_\beta$ results in $D_{KL}^{max} \approx \epsilon_{KL}/C_\beta < \epsilon_{KL}$. This difference between the expected convergence in the primary phase ($\epsilon_{KL}/C_\beta$) and the exit condition of the fixup phase ($\epsilon_{KL}$) is effective at limiting the number of iterations of the fixup phase, as we show below.

In practice, $\pi_\theta$ and the value function may share some of the parameters $\theta$, so the loss function on $\theta$ includes a loss $L_{VF}$ on the value function. Typically, $L_{VF}$ will be the mean squared error of the predicted returns, although value clipping may also be used (Andrychowicz et al., 2020), which most PPO implementations use by default. Combining the policy gradient loss with the value function loss and KL penalty results in a comprehensive loss on the neural network parameters that we use in Algorithm 1:

$$L_\theta = L_\pi + L_{VF} + \beta L_{KL} \tag{5}$$

### 3.2 FIXUP PHASE

The most significant unique component of FixPO is the *fixup phase*, which runs after the *primary phase*. In the primary phase (lines 5 - 7 in Algorithm 1), we repeatedly optimize $\gamma_\theta L_\theta + \gamma_\beta L_\beta$. By choosing $\gamma_\beta$ such that $\gamma_\beta >> \gamma_\theta$ (and $L_\theta$ and $L_\beta$ have similar scales), minimizing the weighted combination of the losses results in approximate convergence to an optimum of $L_\beta \approx 0$. However, it is still possible that $D_{KL}^{max}\left[\pi_\theta, \pi_{\theta'}\right] > \epsilon_{KL}$, and thus that the trust region constraint is not satisfied. To guarantee that the trust region is enforced, the fixup phase iterates through all minibatches, and checks to see if the constraint is satisfied at every state. If the constraint is not satisfied at any state, then the fixup phase performs an update using $\gamma_\theta L_{KL} + \gamma_\beta L_\beta$ and resumes checking all minibatches, as described in lines 8 - 15 in Algorithm 1.

**Algorithm 1:** FIXPO

**Data:** Policy $\pi_\theta(a|s)$
**Data:** Value Function $V_\theta(s)$
**Result:** Optimized parameters $\theta^*$

1 **foreach** $i \leftarrow 1$ **to** *n_policy_improvement_steps* **do**
2     $D \leftarrow \texttt{rollout}(\pi_\theta)$
3     $\pi_{\theta'} \leftarrow \pi_\theta$
4     **foreach** $j \leftarrow 1$ **to** *n_epochs* **do**
5        **foreach** $(s, a, \hat{A}) \leftarrow \texttt{minibatch}(D)$ **do**
6           $\theta \leftarrow \theta - \gamma_\theta \nabla L_\theta(s, a, \hat{A})$ using Equation 5
7           $\beta \leftarrow \beta - \gamma_\beta \nabla L_\beta(s, a)$ using Equation 4
8        **repeat**
9           $fixed \leftarrow True$
10           **foreach** $(s, a) \leftarrow \texttt{minibatch}(D)$ **do**
11              **if any** $D_{KL}(\pi_\theta(s), \pi_{\theta'}(s)) > \epsilon_{KL}$ **then**
                // Unset fixed so we re-check every state
12                 $fixed \leftarrow False$
13                 $\theta \leftarrow \theta - \gamma_\theta \nabla \beta L_{KL}(s, a)$ using Equation 3
14                 $\beta \leftarrow \beta - \gamma_\beta \nabla L_\beta(s, a)$ using Equation 4
15        **until** $fixed$

**Fixup Phase Termination**   Because the fixup phase does not terminate until the trust region constraint is satisfied, it is evident that the trust region constraint is enforced between every policy update. Although we cannot guarantee the fixup phase terminates in general, there are strong reasons to expect it to terminate in practical cases. Because $L_{KL} = 0$ when $\theta = \theta'$, we know that a global optima of $L_{KL} = 0$ exists. Therefore, assuming the central result of Kawaguchi (2016) can be extended to this case, all local optima of $L_{KL}$ equal 0 for these policy classes. Consequently, we can expect the fixup phase to terminate as long as it is able to optimize $L_{KL}$ to a local optimum. In theory, convergence to such a local optima may take an arbitrary number of gradient steps, and require an arbitrarily low learning rate. By applying an upper bound to $\beta$ and decreasing $\gamma_\theta$ when $L_{KL}$ reaches a plateau, $\theta$ such that $D_{KL} < \epsilon_{KL}$ can be guaranteed to eventually be found, although without any upper bound on the runtime. In practice, using $C_\beta > 1$ requires $L_{KL}$ to only be optimized to near a local optimum for the trust region constraint to be satisfied, and consequently for sufficiently large $C_\beta$ values the fixup phase only requires a very small number of gradient steps to terminate, as shown in Figure 1.

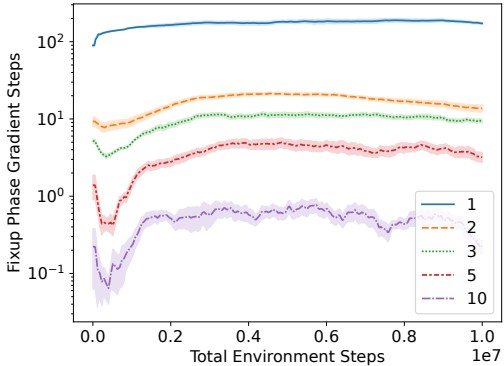

Figure 1: Number of gradient steps performed in the fixup phase throughout training on Walkder2d using different values of $C_\beta$. Larger $C_\beta$ values result in fewer gradient steps but may decrease performance. We found $C_\beta = 3$ to perform well and requires only $5 - 10$ additional gradient steps per policy improvement step, a small increase to the 160 gradient steps performed in the primary phase. The shaded region is the standard error over 10 seeds. See the $C_\beta = 1$ ablation in Figure 5 for details of how reward is negatively affected by a low $C_\beta$.

**Subroutines**   Algorithm 1 makes use of two subroutines, `rollout` and `minibatch`. `rollout` runs full episodes of the policy $\pi_\theta$ in the MDP, collects the resulting tuples, and computes advantages $\hat{A}$ using a value function (also parameterized by $\theta$) and GAE (Schulman et al., 2015b). `minibatch` splits the collected data into minibatches on which we compute loss terms. Except when noted, we use the implementation of these routines typically used in Tianshou (Weng et al., 2022).

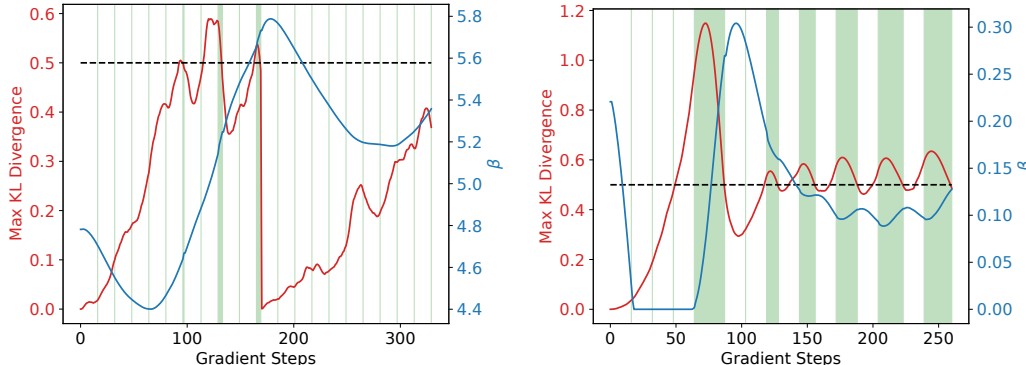

Figure 2: These figures show an example of the interaction between $D_{KL}^{max}(\pi_\theta, \pi_{\theta'})$ (in red) and $\beta$ (in blue) during two consecutive policy improvement steps when $C_\beta = 3$ (left), and during one policy improvement step when $C_\beta = 1$ (right). $L_\beta$ increases $\beta$ when the trust region constraint is violated (the red line is above the dashed line). Solid green regions correspond to gradient steps performed in the fixup phase at the end of each epoch. Vertical green lines show when the fixup phase performed zero gradient steps. Optimizing $L_\beta$ when $C_\beta = 3$ (left) results in $D_{KL}^{max}(\pi_\theta, \pi_{\theta'}) < \epsilon_{KL}$, requiring a few gradient steps in the fixup phase (shown in green), to enforce the trust region. Optimizing $L_\beta$ when $C_\beta = 1$ (right) results in $D_{KL}^{max}(\pi_\theta, \pi_{\theta'}) \approx \epsilon_{KL}$, requiring a large number of gradient steps in the fixup phase to enforce the trust region.

**Using Momentum Optimizers** As is standard practice (Andrychowicz et al., 2020), we use Adam (Kingma & Ba, 2014) (instead of SGD) to optimize both $\theta$ and $\beta$. Therefore, in the initial gradient steps of the fixup phase, optimizing $L_{KL}$ also optimizes $L_\theta$, and optimizing $L_\theta$ in the next few iterations of the primary phase additionally optimizes $L_{KL}$. We have not found this to be a problem in practice using the default hyperparameters for Adam, as long as $C_\beta \geq 2$.

## 4 EXPERIMENTS

### 4.1 GYM MUJOCO CONTROL TASKS

Our first experiments demonstrate that FixPO performs competitively to other trust region methods on the Mujoco control tasks from the OpenAI Gym (Brockman et al., 2016), a finite horizon, continuous action space RL benchmark. We compare our implementation using the Tianshou RL framework (Weng et al., 2022) to the PPO-clip implementation in that framework, as well as to the KL projection layer described in Otto et al. (2021). The results in Figure 3 show that FixPO is generally able to match or exceed other trust region methods on these tasks, and exhibits consistent training stability.

**Hyper-Parameter Robustness** FixPO appears to be highly robust to choices of hyperparameter values. As we will show in Section 4.2, FixPO can perform moderately well with many of its components removed in isolation. We performed hyperparameter sweeps for all of the major hyperparameters, notably $C_\beta, \epsilon_{KL}, \gamma_\beta$, the minibatch size, and the batch size. Changing these parameters within a wide range of values had minimal effect on the algorithm's wall-clock time and rewards. In particular, performance was approximately equal while $0.1 \leq \epsilon_{KL} \leq 0.5$, $2 \leq C_\beta \leq 10$, $0.001 \leq \gamma_\beta \leq 0.1$, and the minibatch size was not more than 512. This allows FixPO to use larger batch and minibatch sizes than the baseline algorithms, allowing for faster wall-clock times in the following experiments. Other experiments we have performed indicate that FixPO does not require the corrections to $\beta$ described in Hilton et al. (2021), which we speculate is due to the constraint on $D_{KL}[\pi_\theta, \pi_{\theta'}]$ more closely following the trust region theory. This includes the Meta-World benchmark, where PPO typically requires batch sizes of at least 50,000 timesteps to stabilize training.

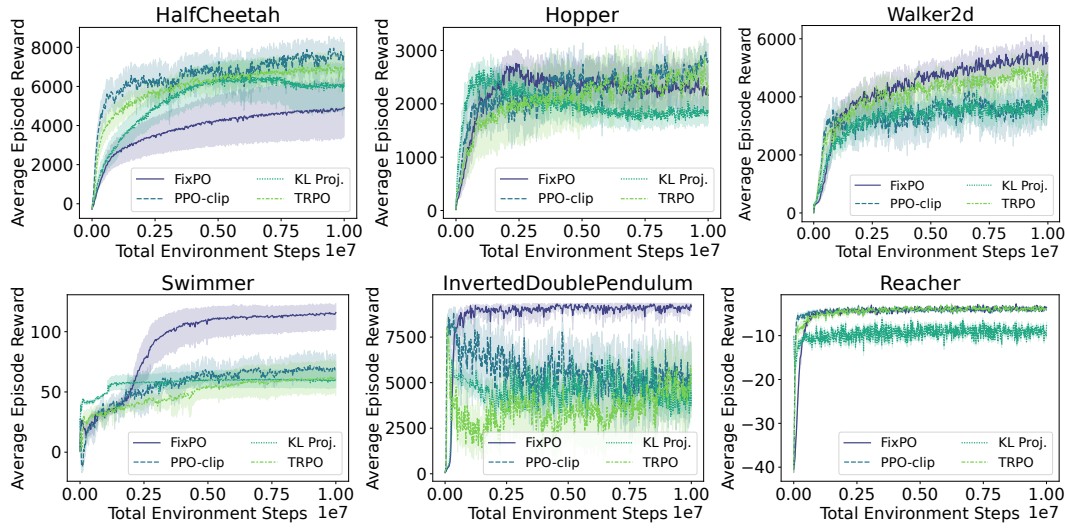

Figure 3: This figure shows the average total reward on the HalfCheetah, Hopper, Walker2d, Swimmer, InvertedDoublePendulum, and Reacher environments as a function of the number of environment steps for FixPO, TRPO, PPO-clip, and the KL projection proposed in Otto et al. (2021). Higher is better. The shaded region is a 95% confidence interval over 10 seeds. FixPO is able to outperform the performance of the other methods on Walker2d, Swimmer, and InvertedDoublePendulum, and consistently avoids large decreases in performance during training. For further analysis on rewards decreasing during training, see Hsu et al. (2020).

**Higher Entropy Policies** On these environments, FixPO naturally learns a higher entropy policy than PPO-clip, without using entropy regularization. This confirms a pattern described in Otto et al. (2021). Figure 4 shows the relative standard deviation of FixPO and PPO-clip on Walker2d.

## 4.2 GYM
### MUJOCO ABLATION EXPERIMENTS

We performed a series of ablation experiments using the Mujoco environments described above. Each ablation experiment removes a unique component of FixPO.

**Remove Fixup Phase** This ablation (labeled `No Fixup Phase` in Figure 5) removes the fixup phase entirely, relying on only tuning $\beta$ in the primary phase to enforce the trust region. This results in an algorithm similar to those described in Andrychowicz et al. (2020). Although

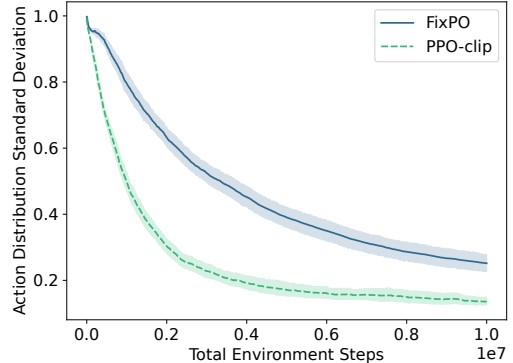

Figure 4: The standard deviation of the action distribution of PPO-clip and FixPO during training on the Walker2d environment. Higher standard deviation corresponds to higher policy entropy, which is known to result in more robust policies (Eysenbach & Levine, 2021), but can produce more variance in performance in the training task, as shown in the HalfCheetah plot in Figure 3. The shaded region is a 95% confidence interval over $\geq 10$ seeds.

this ablation is able to perform well in most runs, we observe poor performance in a portion of runs due to the trust region not being reliably enforced. This matches the theoretical and experimental predictions made of KL regularized PPO in Hsu et al. (2020). Although this ablation achieves higher reward on most tasks than FixPO, it does not guarantee that the trust region is enforced, which is the primary objective of FixPO.

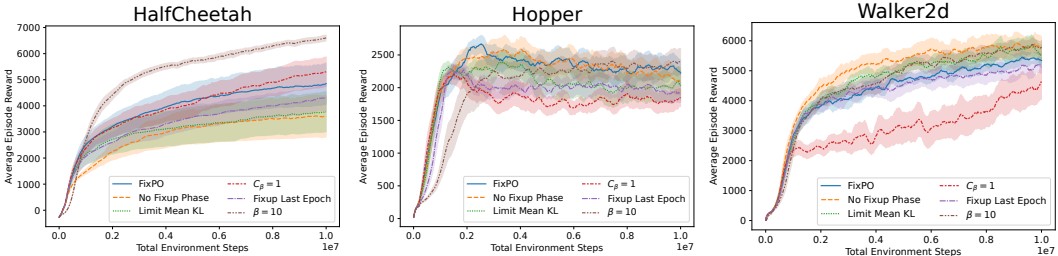

Figure 5: This figure shows the average total reward on the HalfCheetah-v3, Hopper-v3, and Walker2d-v3 environments as a function of the number of environment steps for each of the ablations described in Section 4.2. Higher is better. The shaded region represents one standard error over 10 seeds. Plots have been smoothed with an exponential weighted moving average for legibility.

**Limit the Mean KL Value** This ablation (labeled `Limit Mean KL` in Figure 5) tunes $\beta$ to limit the mean value of the KL divergence, instead of limiting the maximal value using the following loss function:

$$L_\beta = \beta \, \mathbf{sg} \left[ \epsilon_{KL} - C_\beta \operatorname{mean}_s D_{KL} \left[ \pi_\theta, \pi_{\theta'} \right] \right] \tag{6}$$

This is similar to losses described in Song et al. (2019); Andrychowicz et al. (2020) but using $C_\beta$ to adjust the optima. In this ablation, we still run the fixup phase introduced in this work, but exit it once the mean KL divergence is less than the target (i.e. $\overline{D_{KL}(s)} \leq \epsilon_{KL}$). We performed a hyper parameter sweep over $\epsilon_{KL}$ for each environment for the ablation, and found this ablation to perform similarly to our proposed $L_\beta$ given an optimal $\epsilon_{KL}$. Although this ablation is able to reach similar rewards as FixPO, we believe the increased sensitivity to the value of $\epsilon_{KL}$ makes it worse.

**Lagrangian Optima on Constraint Boundary ($C_\beta = 1$)** In this ablation, we remove $C_\beta$ by setting it to 1. This results in the optima of $L_\beta + L_\theta$ being a $\theta$ such that $D_{KL}^{max} \left[ \pi_\theta, \pi_{\theta'} \right] \approx \epsilon_{KL}$. Due to the optima not being reached exactly, and bias introduced by minibatching the losses, it is often the case that $D_{KL}^{max} \left[ \pi_\theta, \pi_{\theta'} \right]$ is significantly greater than $\epsilon_{KL}$, requiring over 100 gradient steps in the fixup phase to correct and significant wall-clock time. A large number of gradient steps in the fixup phase also appears to result in poor reward, which we attribute to catastrophic forgetting in the value function network. See Figure 1 for the effect of this ablation on the number of gradient steps in the fixup phase.

**Only Run Fixup Phase in Last Epoch** In this ablation, we move the fixup phase out of the epoch loop and run it only between policy improvement steps after all epochs have been completed. Equivalently, we run the fixup phase only on the last epoch of each policy improvement step. Due to only needing to enforce the trust region between each policy update step (and not each epoch), this ablation still enforces the trust region guarantee. This results in performing a smaller number of fixup phase gradient steps. However, we found that this ablation slightly decreased the rewards on most environments, including all shown here. Decreasing the overall number of gradient steps by $< 5\%$ also did not measurably improve wall clock time. If decreasing fixup phase gradient steps is absolutely necessary, increasing $C_\beta$ is more effective than this ablation.

**Use a Constant $\beta = 10$** In these experiments, we do not use $L_\beta$, and use a constant value of $\beta = 10$. The fixup phase is still performed. Equivalently, in these experiments $\gamma_\beta = 0$. This ablation performs very well on HalfCheetah, and moderately well on other environments. However, in some individual runs a large number of gradient steps are necessary in the fixup loop, and rewards sometimes decrease significantly. Notably, we were only able to perform this ablation because we observed that $L_\beta$ often tuned $\beta$ to a value between 5 and 20. We believe that the decrease in stability and the need to potentially tune $\beta$ make this ablation worse than the proposed method.

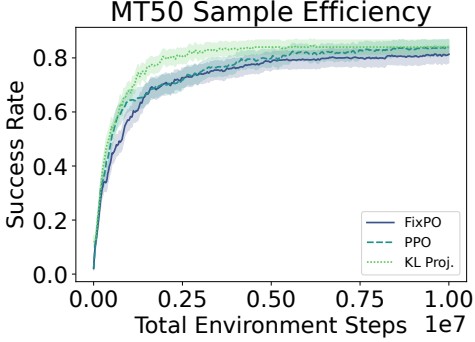
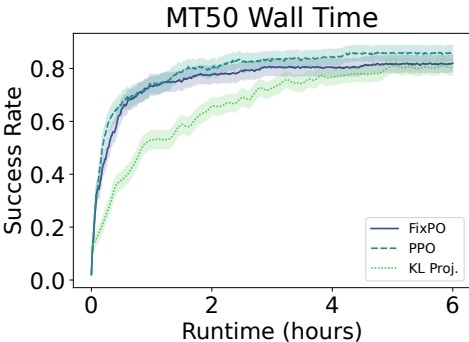

Figure 6: In these experiments we ran 3 separate seeds for each of the 50 v2 tasks in MT50 (with randomized per-episode goals), for each of three algorithms: FixPO, the KL projection from Otto et al. (2021), and PPO Schulman et al. (2017b). All three plots show the average success rates of the 150 runs per algorithm as an aggregate. On the left we show the average success rate during training vs. the number of environment steps per run, with the uncertainty as standard error. All algorithms perform similarly, although Otto et al. (2021) is slightly more sample efficient early in training. In the right plot we show the average success rate as a function during training vs. the number of hours spent training. Here we can see the computational overhead of the optimization used in Otto et al. (2021), although performance between algorithms is similar after six hours.

## 4.3 META-WORLD EXPERIMENTS

In this section, we use the Meta-World Yu et al. (2019), a continuous action space infinite horizon RL benchmark, to run a very large number of experiments comparing FixPO to other trust region methods. Due to these tasks containing randomized goals and starting states, PPO-clip requires a very large batch size (50000), to solve these tasks, or suffers from high instability when trainig. In Figure 6, we use 450 experiment runs to demonstrate that FixPO is able to match the performance of other trust region methods, without requiring a change in hyper parameters. In Figure 7, we perform some simple Transfer RL experiments that show how FixPO is able to finetune without any special handling of the value function, such as described in Zentner et al. (2022).

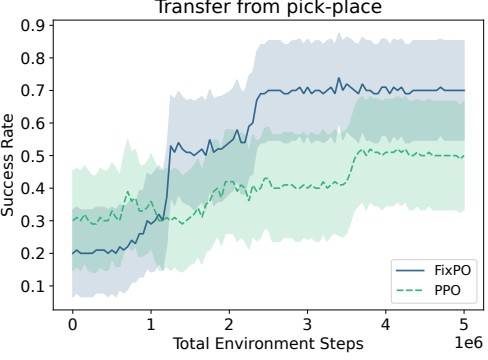
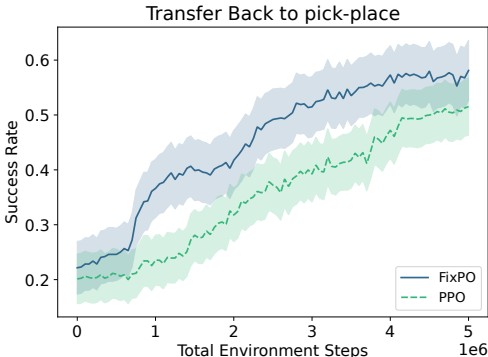

Figure 7: In these figures we show the results of some basic transfer learning experiments using the Meta-World MT10 benchmark. For each algorithm, we pre-train a policy on the `pick-place` task, with randomized goal locations. Then, on the left, we show the success rate of fine-tuning that pre-trained policy aggregated across all 10 tasks in MT10. Following this first finetuning, we then finetune the policy back to the original `pick-place` task. In both cases, FixPO is able to achieve a higher success rate than PPO-clip, and is able to effectively transfer without any additional modifications. Shaded area is standard error over $\geq 10$ runs.

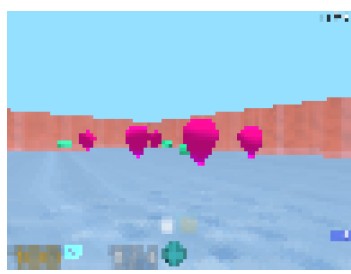 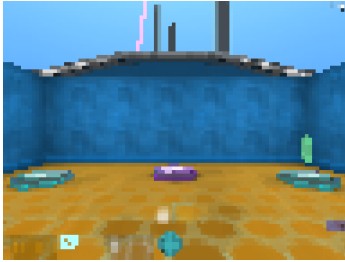 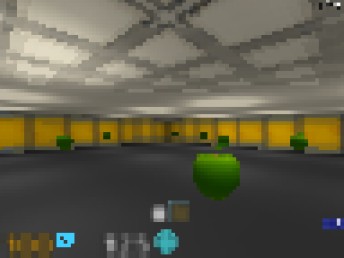

Collect Good Objects          Select Nonmatching Object          Exploit Deferred Effects

Figure 8: Screenshots of three DMLab-30 used (Rooms Collect Good Objects Train, Rooms Select Nonmatching Object, and Rooms Exploit Deferred Effects Train).

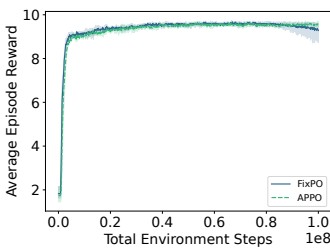 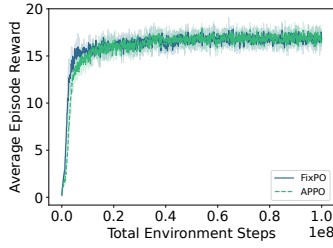 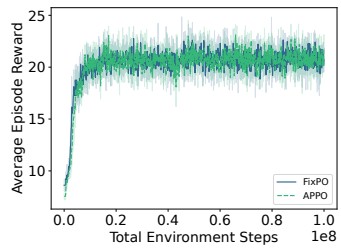

Figure 9: Average episode rewards of FixPO and APPO on the tasks shown above. The performance of FixPO and APPO is approximately equal in these tasks, and we are able to run FixPO for $> 100M$ timesteps. The shaded region is the 95% confidence bounds across 4 seeds.

### 4.4 DMLab-30 Experiments

To demonstrate that FixPO is able to scale where constrained optimization (specifically Schulman et al. (2015a)) cannot, we implement FixPO in the popular `sample-factory` RL framework. This implementation of FixPO was based on the highly performant implementation of APPO in `sample-factory`. We chose the DMLab-30 Beattie et al. (2016) environment because of its high dimensional visual observation space and partial observability, both properties that make training policies with constrained optimization challenging due to the large number of neural network parameters required.

We compared the reward of FixPO and APPO on three DMLab-30 tasks: rooms collect good objects train, rooms select nonmatching object, and rooms exploit deferred effects train. To make the comparison fair, FixPO uses the same hyperparameters as APPO, except for hyperparameters specific to FixPO, which we set $\epsilon_{KL} = 1.0$ and $C_\beta = 2$. Figure 9 shows that FixPO is able to match the performance of APPO on those tasks.

## 5 Limitations

**Multi-Task RL** We experimented with running FixPO as a multi-task RL method on the DMLab-30 benchmark. However, we found that strictly applying the trust region constraint across all tasks simultaneously prevented progress from being made on multiple tasks at once. In the future, we would like to experiment with using one $\beta$ value per task, which may alleviate this limitation.

**More Policy Architectures** One of the advantages of FixPO relative to prior trust region methods is the ability to combine minibatching with trust-region optimization of policies besides Gaussian policies (which works such as Otto et al. (2021) are limited to). Our DMLab-30 experiments show these capabilities in a discrete action space, and we were also able to run our implementation using the Tianshou framework on the Arcade Learning Environment Bellemare et al. (2013). However, further experiments with different action distributions would be a useful direction for future work.

## 6 Conclusion

In this work we have shown how FixPO is able to combine the guarantees of trust region methods with the computational efficiency and rewards of proximal methods. FixPO enforces its trust region via KL penalization, which is flexible and well understood in the machine learning community. In future work, we would like to extend our work to a multi-task setting.

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
