# OpenReview forum: "Guaranteed Trust Region Optimization via Two-Phase KL Penalization"
_ICLR.cc/2024/Conference — Submitted to ICLR 2024_

### Official Review · Reviewer_cJrU · 2023-10-23

**Soundness:** 3 good
**Presentation:** 2 fair
**Contribution:** 2 fair
**Rating:** 5
**Confidence:** 4

**Summary:**

The paper proposes an algorithm FixPO that enforces the trust region constraint to be strictly satisfied during policy learning. Specifically, after the policy update with KL-regularized policy loss, FixPO adds a fixup phase to restrict the KL divergence between new and last policies on each minibatch. The authors compare their method with previous PPO methods on mujoco tasks and metaworld benchmark.

**Strengths:**

- The experiment of this paper is extensive. The authors test the method on different domains and conduct comprehensive ablation study.
- The proposed method exceeds the baselines on several mujoco tasks (e.g., walker2d, swimmer, pendulum) and metaworld transfer learning tasks.
- The additional computation overhead of FixPO is small, which makes it an efficient plug-in component for existing baselines.

**Weaknesses:**

- The authors propose to add a new fixup phase to make the updated policy strictly satisfy trust region constraint, which can reduce the instability during training. However, we can easily address the instability issue by increasing the coefficient of KL regularization $\beta$, which is not only easier to implement, but also achieves better performances according to the ablation study (fig.5, $\beta=10$).
- The advantage of FixPO over previous methods is not consistent. In mujoco domain, FixPo does not exhibit better performance than baselines on halfcheetah, hopper and reacher tasks. There is also no significant different between FixPO and APPO in DMLab tasks. Meanwhile, FixPO cannot even exceed No Fixup phase baseline in fig.5, which makes the effectiveness of fixup phase very questionable.

Minor issues:
- The line below eq.(1), $L_{\pi_i}$ should be "policy loss" instead of "policy gradient".
- The font of pdf is inconsistent with the given ICLR template.

**Questions:**

- What is the exact definition of $D_{KL}^{max}[\pi_1,\pi_2]$?

---

> ### Author Response · Authors · 2023-11-17
>
> Thank you for the detailed review.
>
> ### Weaknesses
> It is true that setting $\beta = 10$ performs much better than expected, especially on the HalfCheetah benchmark. However, this does not guarantee the trust region is enforced without also using the fixup phase, which that ablation does use, since even $\beta = 10$ may not be sufficient to enforce the trust region. In some runs of other environments, $\beta$ reached much higher values than 10.
>
> > The advantage of FixPO over previous methods is not consistent.
>
> We do not claim that FixPO consistently achieves higher rewards than prior works. Our experiments are intended to demonstrate that FixPO is able to approximately match the rewards of proximal methods while providing stronger guarantees, and is able to match the guarantees of prior trust region methods while being more computationally efficient than them.
>
> > The line below eq.(1), $L_{\pi_i}$ should be “policy loss” instead of “policy gradient”.
>
> $L_{\pi_i}$ is the “importance sampled policy gradient loss”, that is described in Section 5 of [1]. We have edited the description to make this more clear.
>
> > The font of pdf is inconsistent with the given ICLR template.
>
> Thank you for bringing the font inconsistency to our attention, we have fixed it in the latest revision.
>
> > What is the exact definition of $D_{KL}^{max}[\pi_1, \pi_2]$?
>
> The exact definition of $D_{KL}^{max}[\pi_1, \pi_2]$ is $max_{s \in D} D_{KL}[\pi_1(a|s) || \pi_2(a|s)] = max_{s \in D} \int_a \pi_1(a|s) \frac {\pi_1(a|s)} {\pi_2(a|s)} \mathrm{d}a$
>
> ### Citations
>
> [1] John Schulman, Sergey Levine, Philipp Moritz, Michael I. Jordan, and Pieter Abbeel. Trust
> region policy optimization. CoRR, abs/1502.05477, 2015.

---

### Official Review · Reviewer_MD36 · 2023-10-30

**Soundness:** 4 excellent
**Presentation:** 4 excellent
**Contribution:** 4 excellent
**Rating:** 10
**Confidence:** 5

**Summary:**

The paper sets out to combine the strengths of PPO and TRPO, namely, efficiency and ease of implementation of PPO and theoretical guarantees of TRPO. To this end, the paper begins with the PPO algorithm (defined in eq. 2) and makes two important modifications. First, the Lagrange multiplier that controls the importance of the KL term in the loss function is learned via dual gradient descent. Second, an additional fixup phase is added after each policy update, which ensures the satisfaction of the trust region constraints by making a few additional gradient steps. The paper highlights the strengths of the proposed algorithm in simulated experiments.

**Strengths:**

- The research question is very important. The dichotomy between the guarantees of TRPO and the simplicity of PPO has been a longstanding issue in RL.
- Every design choice is well-motivated and complimented with experiments. At the same time, the algorithm is simple to implement and does not add much time compared to PPO.
- The paper is extremely clear, concise, and well-written. Most questions I had while reading it (e.g., how come the trust region is guaranteed, what if the fixup phase did not terminate, or what if we removed the fixup phase) were answered later in the paper.
- The experiments are extensive and diverse. Many relevant environments and ablations are included.

**Weaknesses:**

- TRPO is a relevant baseline but is absent from experiments.

**Questions:**

- Shouldn’t $D_{KL}$ be multiplied by $C_\beta$ in line 11 of Algorithm 1 (according to eq. 4)?
- Is the fixup phase analogous to line search in TRPO?
- Can an entropy term be added to encourage exploration? I understand the point made in Fig. 4 but still wonder if additional exploration could be beneficial.
- I find it interesting that in HalfCheetah, FixPO underperforms both PPO (fig. 3) and the constant $\beta$ ablation (fig. 5). I wonder if using a high constant $\beta$ (maybe without fixup) results in approximate trust region similar to what clipping does in PPO.
- Just in case the authors would appreciate more related work on the Lagrangian optimization (where the Lagrange multiplier is learned), similar approaches are used in Adversarial (Inverse RL / Imitation Learning / Generative Networks) (https://arxiv.org/abs/1810.00821), Differentiable Economics (https://arxiv.org/abs/1706.03459, https://arxiv.org/abs/2202.13110), and Multi-Agent RL (https://arxiv.org/abs/2306.08419).

---

> ### Author Response · Authors · 2023-11-17
>
> Thank you for your kind and thoughtful comments.
>
> We have revised the paper to include a TRPO baseline. It performs similarly to other baselines, so it does not significantly affect the interpretation of the results.
>
> ### Questions:
>
> > Shouldn’t $D_{KL}$ be multiplied by $C_\beta$ in line 11 of Algorithm 1?
>
> This is a subtle but important aspect of the algorithm. By using $C_\beta$ in equation 4 and not using it on line 11, we effectively optimize a more conservative trust region in the primary phase minibatch steps than in the fixup phase minibatch steps.
> This is important for two reasons.
> First, because if the primary phase optimizes so that $D_{KL}^{max}$ is approximately at the trust-region boundary, it will often be slightly outside the trust-region boundary, requiring fixup gradient steps to correct it.
> Second, because the minibatch is likely to produce a smaller $D_{KL}^{max}$ than the entire batch, we use $C_\beta$ in the primary phase minibatches so the KL computed there approximately matches the trust region constraint checked in the fixup loop.
>
> If we were to use $C_\beta$ on line 11, it would be mathematically equivalent to changing the hyper parameters so that $\epsilon_{KL} \leftarrow \epsilon_{KL} / C_\beta$, $\gamma_\beta \leftarrow \gamma_\beta / C_\beta$, and $C_\beta \leftarrow 1$.
> However, setting $C_\beta = 1$ performs very badly, as shown in the ablation.
> In comparison, the extra conservativeness of the minibatch trust region does not seem to have a negative effect on reward, probably because the exact size of the trust region was largely chosen arbitrarily.
>
> We have revised the paragraph after equation 4 to make this more clear.
>
> > Is the fixup phase analogous to line search in TRPO?
>
> The fixup phase performs a similar function to line search in TRPO, and in that sense they are definitely analogous. There are also several important differences. The most significant difference is that the fixup phase continues to use minibatches of gradient steps (SGD), while TRPO line search uses a projected gradient of the entire batch. For some environments, such as DMLab-30, the entire batch is very large (>50k samples), and consequently it is impractical to compute the full batch projected gradient.
> The fixup phase also differs from the line search in TRPO through how the initial value step size is calculated. TRPO calculates a starting step size using the Hessian vector product, then shrinks that step using an exponential line search until the trust region is satisfied. FixPO checks to see if any gradient steps are necessary, and takes stochastic gradient steps to “decrease the step size” until the constraint is satisfied.
>
> > Can an entropy term be added to encourage exploration?
>
> An entropy term can definitely be added to $L_\theta$. We briefly experimented with doing so, but did not use it in the final version so that we could focus more on other details in Section 3.
>
> > Question about HalfCheetah
>
> We were surprised by the high performance of the $\beta = 10$ ablation on the HalfCheetah environment. When using $\beta = 10$ on HalfCheetah, the KL divergence remains very low and almost no fixup steps are required. We suspect that HalfCheetah benefits from having a smaller trust region than the other environments, but did not want to tune hyper parameters separately for different environments.
>
> > Additional citations for Lagrangian Methods
>
> Thank you very much for providing us with these citations, we have incorporated them into the paper.

---

> > ### Comment · Reviewer_MD36 · 2023-11-17
> > **Post-rebuttal**
> >
> > Thank you for answering my questions, makes sense to me.
> >
> > I also read other reviews and their rebuttals. My assessment remains unchanged.

---

> ### Comment · Area_Chair_2EU5 · 2023-11-22
> **Justification of "10"**
>
> Dear Reviewer,
>
> Could you read the other reviews and tell me why you think it is worth "10" (the perfect score)? A short review with "10" (given there are "3" from the other reviewer) is not informative and convincing.
>
> Best,
> AC

---

> > ### Comment · Reviewer_MD36 · 2023-11-22
> > **Ok sure**
> >
> > I disagree that my review is uninformative, especially compared to some other reviews for this very same paper. I incline you to ask them this same question: "Could you read the other reviews and tell me why you think it is worth "3"? A short review with "3" (given there are "10" from the other reviewer) is not informative and convincing"
> >
> > Regardless, I will be happy to elaborate.
> >
> > **My review**
> >
> > Let's go over my assessment for each criterion.
> >
> > *Presentation*
> >
> > "The paper is extremely clear, concise, and well-written.". That is, the relevant points are covered in a short and precise manner, and at the same time, what is covered is relevant.
> >
> > The introduction covers the background on TRPO and PPO, and in particular, the guarantees of the former vs the simplicity of the latter. Then, the contribution is stated as a method that is best of the both worlds.
> >
> > The relevant work is short but on point. It not only covers trust region methods (which is a must-have) but also Lagrangian and KL-regularized methods (which is a more broadly related work) and even has a separate paragraph about libraries (which is convenient for reproducibility).
> >
> > The method has several innovations that are introduced in a logical order and are clearly explained.
> >
> > The experiments are the largest section, which is the right decision. These include comparisons to relevant benchmarks and ablations (testing the aforementioned innovations) in Mujoco, Meta-World, and DMLab-30 (I would be fine with 2 out of these 3 benchmarks).
> >
> > A separate section for limitations is commendable.
> >
> > The few points I did not fully understand, the authors clarified in the rebuttal and the revision.
> >
> > *Contribution*
> >
> > The contribution is a method that is 1) simpler than other trust-region methods that provide guarantees and 2) provides guarantees in contrast to other simple (proximal) methods. This might seem somewhat incremental as the performance is only on par with PPO (and who cares about guarantees if the curve do not go up), but I think this is important because of the wide applicability of PPO /TRPO in all domains of RL. Thus, a study with such a contribution could be relevant to a broad RL auditory.
> >
> > *Soundness*
> >
> > The study is empirical; the guarantees come from the design of the method rather than the formal proofs. So to assess soundness, we need to look closely at the experiments. On a high level, the experiments should be designed to test all claims. That is, to verify the main contribution, as well as every detail of the method explained in the Method section. The experiments should also cover relevant baseline algorithms and benchmark environments. The methods should be compared fairly. To my understanding, all these marks are checked: there are a lot of environments; relevant baselines are provided (TRPO and PPO are sufficient), using standard implementations; ablations verify the necessity of implementation details and robustness to hyperparameters. If I had any nitpicks here, confidence intervals are provided but not statistical tests, but to me, standard errors are enough.
> >
> > **Other reviews**
> >
> > *HunE*
> >
> > The weaknesses of the presentation come from some mixture of failure in reading comprehension (such as claiming that $L_\beta$ is not used) and unfamiliarity with RL (such as having questions about what advantage is or how value function is trained).
> >
> > The two comments about soundness are valid, but I was satisfied with the authors' response. In particular, the reviewer points out that PPO achieved higher performance in some MuJoCo environments in the original paper, to which the authors reply that 1) the training time differs and 2) other papers have also encountered these discrepancies. I wonder what the reviewer's thoughts are on this, but they did not engage in a conversation. From my side, I can add that I'm not surprised that performance in MuJoCo has been inconsistent over the years. Back in the day, there were two versions of mujoco -- one paid and one open-source (pybullet). Now the paid version changed hands, became free, and integrated with gym. The backend may have been rewritten or the parameters changed.
> >
> > The concern about incremental contribution I have already disagreed with before.
> >
> > *bxPw*
> >
> > The reviewer did not understand the paper and asked for clarifications, which is fine as long as there is a further effort to reassess the paper during the discussion period. Again, this was not done.
> >
> > *cJrU*
> >
> > I think the core disagreement is again on the relevancy of the contribution, which I already discussed.
> >
> > **Summary**
> >
> > I hope this has been more informative and/or convincing. I have not taken the time to enumerate everything I like about the paper and every comment of other reviewers I disagree with because, well, it takes time. But as you can see, I am well prepared to defend my assessment.

---

> ### Comment · Reviewer_MD36 · 2023-11-22
> **Ok sure PS**
>
> **Notes on the scoring system**
>
> In the new scoring system at ICLR, I had little room to differentiate between 'accept' and 'strong accept'. In some other conferences, "strong accept" is an 8/10, and 10/10 is reserved for "this is the best paper I've ever seen". In this other system, maybe I would've given an 8/10. Maybe I would've bumped it to 9 after reading other reviews and after the authors addressed my TRPO concern. At this conference, 10/10 is "strong accept", which I read as including "the best paper I've ever seen", but also including just "strong accept". 8/10 is "accept" and there is no inbetween. I'm ok with this system as it removes some noise, but evidently, not all ACs are accustomed to it.

---

### Official Review · Reviewer_bxPw · 2023-11-05

**Soundness:** 3 good
**Presentation:** 3 good
**Contribution:** 3 good
**Rating:** 6
**Confidence:** 2

**Summary:**

This paper proposes an adaptive strategy for improving the performance of trust region optimization for solving RL problems. In particular, the proposed algorithm can adaptively fine tune the parameter beta so that the KL divergence of two consecutive policies can within the user-specified distance, without putting this requirement as a hard constraint.

**Strengths:**

I think this paper provides practical approach to efficiently solve RL problems. To the best of my knowledge, this paper is novel. The proposed algorithm is validated in many test instances, which is very nice.

**Weaknesses:**

In general I think the author(s) explain the high level idea of the proposed method very well. However, I do not totally understand the intuition behind equation (4), which is perhaps the most important step in the proposed algorithm. Why it works and controls the distance between pi and pi'? It would be great if this result could be written as a proposition with proof.

Also it would be great if the author(s) could provide the per-iteration complexity of the proposed method, compared to the other state-of-the-art approaches.

**Questions:**

Could you mathematical describe the reasoning behind equation (4)?

How do you compute the gradients of L_theta and L_beta?

What is the complexity of the fixed up phase?

---

> ### Author Response · Authors · 2023-11-17
>
> Thank you for your review.
>
> Because you asked questions for each weakness mentioned, we will respond to the questions directly.
>
> ### Questions
>
> > Could you describe the mathematical reasoning behind equation 4?
>
> As you mention, equation 4 is one of the most important steps in the proposed algorithm. This equation results in $\beta$ acting as a Lagrange multiplier, which is a methodology also employed in prior works mentioned in the “Lagrangian Methods” paragraph of the related work section. An alternative explanation is to see $L_\beta$ as applying a linear feedback controller to the loss that approximately enforces the trust region constraint on a per minibatch basis. The use of momentum optimization for $\beta$ then causes this controller to be a PI (Proportional & Integral) controller. From this perspective, $C_\beta$ translates between a per-policy-step target $\epsilon_{KL}$ to a per-minibatch-step target $\epsilon_{KL} / C_\beta$.
> We describe why we expect this to result in convergence to the per-minibatch-step target in Section 3.2, but unfortunately we were unable to find a proof of this convergence in all cases.
>
> > How do you compute the gradients on $L_\theta$ and $L_\beta$?
>
> We compute gradients of our losses using the back propagation algorithm, as implemented in PyTorch.
>
> > What is the complexity of the fixup phase?
>
> Unfortunately, proving an upper bound on the per-iteration complexity of the proposed method would involve proving a finite-time convergence proof for SGD applied to deep neural networks, which is a major open problem in deep learning theory.
> We cite the most recent progress we are aware of on this open problem, [1], in Section 3.2.
> In practice, the average time spent in the fixup phase is quite low, as shown in Figure 1, and appears to be roughly constant throughout training for a specific choice of hyper parameters.
> We do observe that occasionally a specific policy update out of several hundred can take significantly longer than others, however this never added more than 10 minutes to a 3 hour experiment run. As an optimization, it may be worth backtracking to a prior $\theta$ value if the fixup phase exceeds a time limit, but we found this to be unnecessary in practice.
>
> ### Citations
>
> [1] Kenji Kawaguchi. Deep learning without poor local minima, 2016.

---

### Official Review · Reviewer_HunE · 2023-11-07

**Soundness:** 2 fair
**Presentation:** 1 poor
**Contribution:** 2 fair
**Rating:** 3
**Confidence:** 3

**Summary:**

The paper proposes a novel trust region policy optimization method that follows the line of TRPO and PPO.

**Strengths:**

The paper proposes a novel policy optimization algorithm testing the proposed approach on different domains.

The idea seems novel, although incremental.

**Weaknesses:**

### Presentation

- The paper does not provide context on the setting considered. Preliminaries are completely missing, the general tone is too colloquial, it is not clear which RL setting is considered (finite-horizon, infinite-horizon, average reward etc. )

- The advantage function $\hat{A}$ is never defined.

- $L_\beta$ is introduced and not used later.

- What is $L_{VF}$?

- How is it computed $L_\pi$?

- Figure 1: How can we extrapolate from the figure that the performances are reduced?

### Experimental evaluation

- The results of PPO-clip on Mujoco control tasks domain are not coherent with the original paper [1].

     - Why are your results on Inverted Pendulum PPO-clip ~5000 when in [1] they achieved ~8000?

     - Why are your results on Swimmer PPO-clip ~50 when in [1] they achieved ~100?

- The comparison with TRPO is missing (as with other policy optimization methods), although the two methods are quite similar.

- In general, the experimental evaluation is not convincing since the proposed method does not provide better results compared to PPO and the comparison with other policy optimization algorithms is missing.

**Questions:**

See weaknesses.

---

> ### Author Response · Authors · 2023-11-17
>
> Thank you for your detailed review. While our work may appear to be an incremental improvement, we believe we propose the first RL method to guarantee a trust region with minimal computational overhead relative to proximal methods.
>
> ### Presentation
> > The paper does not provide context on the setting considered … it is not clear which RL setting is considered
>
> We have added a clarification to the description of each benchmark describing the RL setting for that benchmark. The Mujoco and DMLab-30 benchmarks are finite-horizon with discounted rewards and the Meta-World benchmark is infinite–horizon and uses discounted rewards with a success rate metric. These are the recommended settings for these benchmarks as used in prior work.
>
> > the general tone is too colloquial
>
> We appreciate the feedback that the tone is not formal enough. We have significantly revised the introduction and ablation sections to use more formal language and include more details.
>
> > The advantage function is never defined.
>
> We compute $\hat{A}$ using GAE[1]. We have added a citation to the second paragraph in Section 3.1 and to the Subroutines paragraph to clarify how $\hat{A}$ is computed.
>
> > $L_\beta$ is introduced and not used later.
>
> We apologize, but we believe you are mistaken about $L_\beta$ not being used. $L_\beta$ is used in two important places (line 7 and line 14) in Algorithm 1. We also describe in detail how $L_\beta$ is used in Section 3.2.
>
> > What is $L_{VF}$?
>
> $L_{VF}$ is described in the last paragraph of Section 3.1. We have not given a formal description of value function clipping because it is described in detail in prior works such as [2].
>
> > How is it computed $L_\pi$?
>
> $L_\pi$ is described in Equation 3 to be the importance sampled policy gradient, $-\frac {\pi_{\theta}(a|s)} {\pi_{\theta'}(a|s)} \hat{A}$. We have added a note to clarify that this description is the definition.
>
> > Figure 1: How can we extrapolate from the figure that the performances are reduced?
>
> Figure 1 is not intended to show the rewards the policy achieves for different values of $C_\beta$, and instead is intended to show stability in the number of gradient steps required throughout training. Figure 5 compares $C_\beta = 1$ to the default we chose ($C_\beta = 3$), which shows that $C_\beta = 1$ results in significantly decreased rewards. We have added a note to the description of Figure 1 referring to the $C_\beta = 1$ ablation. Figure 1 does show that $C_\beta = 1$ results in taking significantly more gradient steps. Gradient steps must be performed in sequence, and consequently taking a large number of them decreases the computational performance of Algorithm 1.
>
> ### Experimental Results
>
> > Why are your results on Inverted Pendulum PPO-clip ~5000 when in [1] they achieved ~8000?
>
> We assume you mean Inverted Double Pendulum. The original PPO paper only reports rewards for InvertedDoublePendulum up to 1 million timesteps. We show PPO achieving similar performance (~8000) at that time in Figure 3. Continuing to train PPO-clip can lead to decreased rewards when not combined with KL penalization, as described in [5].
>
> > Why are your results on Swimmer PPO-clip ~50 when in [1] they achieved ~100?
>
> We were not able to reproduce the Swimmer results reported in the original PPO paper. Results on Swimmer are missing from most papers using PPO, including [3] and [4]. Where rewards on Simmer are reported in [5], they match our results of ~50.
>
> > The comparison with TRPO is missing (as with other policy optimization methods), although the two methods are quite similar.
>
> We have added a comparison to TRPO for OpenAI Gym. It is computationally difficult to add results for TRPO on Meta-World within the time limit of the revisions period, but we currently compare FixPO to [4] on this benchmark, which was published at ICLR and is reported to perform better than TRPO.
>
> > the proposed method does not provide better results compared to PPO and the comparison with other policy optimization algorithms is missing.
>
> The primary objective of this work is not to achieve higher rewards than PPO, our experimental results are intended to show that we can maintain the same peak reward at similar computational cost while providing stronger guarantees.
>
> ### Citations to follow in separate comment.

---

> > ### Author Response · Authors · 2023-11-17
> > **Citations for preceding comment**
> >
> > [1] John Schulman, Philipp Moritz, Sergey Levine, Michael I. Jordan, and Pieter Abbeel. High-dimensional continuous control using generalized advantage estimation. CoRR, abs/1506.02438, 2015.
> >
> > [2] Engstrom, L., Ilyas, A., Santurkar, S., Tsipras, D., Janoos, F., Rudolph, L., & Madry, A. (2020). Implementation Matters in Deep RL: A Case Study on PPO and TRPO. International Conference on Learning Representations. Retrieved from https://openreview.net/forum?id=r1etN1rtPB
> >
> > [3] Huang, et al., "The 37 Implementation Details of Proximal Policy Optimization", ICLR Blog Track, 2022. Retrieved from https://iclr-blog-track.github.io/2022/03/25/ppo-implementation-details/
> >
> > [4] Otto, F., Becker, P., Vien, N. A., Ziesche, H. C., & Neumann, G. (2021). Differentiable Trust Region Layers for Deep Reinforcement Learning. arXiv [Cs.LG]. Retrieved from http://arxiv.org/abs/2101.09207
> >
> > [5] Hsu, C. C.-Y., Mendler-Dünner, C., & Hardt, M. (2020). Revisiting Design Choices in Proximal Policy Optimization. arXiv [Cs.LG]. Retrieved from http://arxiv.org/abs/2009.10897

---

### Author Response · Authors · 2023-11-17
**We have posted a revision with formatting fixes, more formal language, and a TRPO baseline**

We would like to thank all reviewers for their helpful feedback. We have uploaded a revision incorporating feedback from each reviewer. Fixing a font inconsistency pointed out by Reviewer cJrU allowed us more writing space, which we used to fix a margin problem on Page 9, to add more precise details to the ablation descriptions in Section 4.2, and to improve the formality of the introduction. As requested by multiple reviewers, we have added a TRPO baseline to the Mujoco experiments. We have also updated the citation style to use name-year citations as described in the ICLR example document. Finally, we have added a variety of small clarifications and layout edits, most of which are mentioned in individual comments below.

---

### Meta-Review · Area_Chair_2EU5 · 2023-12-12

**Metareview:**

The paper proposes an algorithm FixPO that enforces the trust region constraint to be strictly satisfied during policy learning.

The reviews are split. As pointed out by Reviewer MD36, the algorithm in this paper is simple and guaranteed.  On the other hand, Reviewer cJrU argues that minor violations of the trust-region constraint can be acceptable if they do not compromise the overall stability and performance of the training process. A critical point of concern is that the improvements offered by FixPO over existing methods are not substantial, which questions the overall contribution of this work.

Given these considerations and the limited advancement presented by the paper, I tend to recommend rejection.

SAC note: I read the paper and the discussions given the disagreement in reviewer scores. Reviewer cJrU raised a critical concern that training stability and performance is the final goal, and the trust region constraint is only a surrogate for achieving such a goal. They pointed out that a fixed large value of $\beta$ often works well, and the authors replied that such $\beta$ is not sufficient for guaranteeing trust-region constraint---this response appears to take the trust-region constraint as the end goal and not directly addressing the reviewer's question. The SAC also agrees with reviewer HunE's criticism that the paper is written too colloquially. For example, p3 mentions that "... is theoretically justified by Equation 2", but the exposition on pages 2--3 does not include any mathematical derivations (including where the Lagrangian multipliers are mentioned), making it difficult to tell which steps follow from derivations and which are heuristic designs.

**Justification For Why Not Higher Score:**

See the above comments for the reason of rejection.

**Justification For Why Not Lower Score:**

N/A

---

### Decision · Program_Chairs · 2024-01-16

Reject